# Supracondylar Fractures of the Humerus: Association of Neurovascular Lesions with Degree of Fracture Displacement in Children—A Retrospective Study

**DOI:** 10.3390/children9030308

**Published:** 2022-02-24

**Authors:** Ryszard Tomaszewski, Karol Pethe, Jacek Kler, Erich Rutz, Johannes Mayr, Jerzy Dajka

**Affiliations:** 1Department of Pediatric Traumatology and Orthopedics, Upper Silesian Child Centre, 40-752 Katowice, Poland; tomaszewskir@gmail.com (R.T.); karol.pethe@gmail.com (K.P.); jacek.kler@gmail.com (J.K.); 2Institute of Biomedical Engineering, Faculty of Science and Technology, University of Silesia, 40-007 Katowice, Poland; 3Department of Orthopaedics, The Royal Children’s Hospital Melbourne, Melbourne, VIC 3052, Australia; erich_rutz@hotmail.com; 4Murdoch Children’s Research Institute, MCRI, Melbourne, VIC 3052, Australia; 5Department of Pediatric Surgery, University Children’s Hospital Basel, University of Basel, 4031 Basel, Switzerland; 6Institute of Physics, Faculty of Science and Technology, University of Silesia, 40-007 Katowice, Poland; jerzy.dajka@us.edu.pl; 7Silesian Center for Education and Interdisciplinary Research, University of Silesia, 40-007 Katowice, Poland

**Keywords:** supracondylar humerus fractures, treatment, children, vascular injury, neurologic injury

## Abstract

Supracondylar humerus fractures (ScHF) account for 60% of fractures of the elbow region in children. We assessed the relationship between neurovascular complications and the degree of fracture displacement as rated on the basis of modified Gartland classification. Moreover, we aimed to evaluate predisposing factors, e.g., age and gender, and outcomes of neurovascular complications in ScHF. Between 2004 and 2019, we treated 329 patients with ScHF at the Department of Traumatology and Orthopedics of the Upper Silesian Child Centre, Katowice, Poland. Mean age of patients (189 boys and 140 girls) was 7.2 years (Confidence interval: 6.89, 7.45). Undisplaced fractures were treated conservatively with a cast. Displaced fractures were managed by closed reduction and percutaneous Kirschner wire fixation using two pins inserted laterally. We retrospectively assessed the number of neurovascular lesions at baseline and recorded any iatrogenic injury resulting from the surgical intervention. Acute neurovascular lesions occurred in 44 of 329 ScHF patients (13.4%). The incidence of accompanying neurovascular injuries was positively associated with the severity of fracture displacement characterized by Gartland score. Vascular injuries occurred mainly in Gartland type IV ScHF, while nerve lesions occurred in both Gartland type III and IV ScHF. We noted a significantly higher mean Gartland score and mean age at injury in the group of children suffering from neurovascular injuries when compared to those in the group without such injuries (*p* = 0.045 and *p* = 0.04, respectively). We observed no secondary nerve lesions after surgical treatment. For the treatment of ScHF in children, we recommend closed reduction and stabilization of displaced fractures with K-wires inserted percutaneously from the lateral aspect of the upper arm. We advocate vessel exploration in case of absent distal pulses after closed reduction but do not consider primary nerve exploration necessary, unless a complete primary sensomotoric nerve lesion is present.

## 1. Introduction

Supracondylar humerus fracture (ScHF) represents a common bone injury in children, accounting for 60% of elbow fractures in the pediatric population [1]. ScHF mainly affects children below the age of seven years [2]. After the age of seven years, ScHF represents the second most frequent fracture [3]. Male predominance is typical, with a male-to-female ratio of 3:2 [4]. 

Extension-type ScHF caused by falling onto an extended elbow account for 97% to 99% of ScHF [2,5,6]. Left or non-dominant limbs are most often affected [4]. Fractures are typically classified according to Gartland [7]. Four types are distinguished, i.e., type I (non-displaced), type II (displaced, but with the intact posterior cortex), type III (completely displaced, with either posteromedial or posterolateral displacement), and type IV (displaced with multidirectional instability due to circumferential periosteal disruption) [8].

Fractures of Gartland types II, III, and IV are usually managed by closed reduction and surgical stabilization. The preferred surgical technique to stabilize ScHF is K-wire fixation, most commonly by inserting two parallel or diverging K-wires from the lateral side of the humerus [9].

The most frequently observed complications after ScHF comprise axial malalignment (e.g., cubitus varus in a frontal plane and residual hyperextension malalignment in a lateral plane), paresthesia, and superficial pin infections [10].

Terpstra et al. found that the incidence of paresthesia complicating ScHF treated between 9 p.m. and 2 a.m. was higher when compared to the incidence of nerve injuries observed in ScHF treated during office hours [11]. According to their literature review, it appears safe to postpone surgery to office hours, if circumstances are not optimal for acute surgery at nighttime and if there is no medical contraindication [11].

Pavone et al. compared the outcomes of ScHF in a group of children treated with crossed pins to the outcomes in a group of children stabilized by lateral pin constructs. They found a satisfactory outcome with similar results regarding joint function recovery and postoperative complications [10]. 

Severe complications related to ScHF, including nerve and vessel lesions, can be primary, i.e., originating from the initial injury, or secondary as an iatrogenic injury resulting from fracture treatment [3,12,13]. Nerve lesions occur in 5.8% to 14% of children suffering from ScHF [5,6,14,15], and the incidence of vascular injuries ranges from 3.2% to 14.3% [6,16,17,18,19]. For ScHF complicated by neurovascular injuries, urgent reduction and stable fixation are recommended [20]. These fractures should be managed by the close collaboration of pediatric surgeons, neurosurgeons, and vascular surgeons [20,21].

We aimed to determine whether there is an association between the degree of fracture displacement and incidence as well as outcome of accompanying neurovascular lesions. We hypothesized that the incidence of primary neurovascular injuries complicating ScHF in children is unrelated to the degree of fracture displacement as classified by Gartland. Moreover, we aimed to evaluate predisposing factors and outcome of nerve and vessel injuries in ScHF.

## 2. Patients and Methods

### 2.1. Demographic and Baseline Assessments

We retrospectively evaluated 329 patients with ScHF who were treated at our department between January 2004 and July 2019. 

We included children younger than 17 years who suffered acute ScHF and were initially referred to the Department of Pediatric Traumatology and Orthopedics, Upper Silesian Child Centre, Katowice, Poland. 

The exclusion criteria comprised patients older than 16 years at the time of injury, polytraumatized patients, patients suffering from pathological fractures or recurrent fractures of the elbow region, and patients suffering from psychomotor dysfunctions or neuromuscular disorders. We also excluded patients with incomplete medical records or loss to follow-up.

We analyzed patient demographics, i.e., age and sex, and baseline characteristics, i.e., type of fracture according to Gartland, presence of accompanying nerve lesions and/or vascular injuries, and laterality of injury. 

### 2.2. Clinical and Radiographic Assessments

Every ScHF patient admitted to the hospital for fracture treatment was assessed clinically and radiographically at the Accident and Emergency department. Fractures were categorized according to the modified Gartland classification based on X-ray images in anteroposterior (AP) and lateral projections [7]. All patients underwent evaluations of radial and ulnar pulse and capillary refill time as well as forearm and hand innervation.

We used the protocol by Marsh et al. [22] to improve the assessment and documentation of the neurologic status in children with arm fractures. The protocol by Marsh et al. does, however, not include any sensory or vascular assessment and does not document posterior interosseous nerve function [22]. A simple finger play for children was used to guide physicians’ assessments of motor function of median, radial, ulnar, and anterior interosseous nerves.

We grouped patients into those with neurovascular complications (group 1) and those without neurovascular lesions (group 2).

### 2.3. Treatments

Non-displaced fractures (Gartland type I) were treated conservatively. Displaced fractures were managed by closed reduction and K-wire fixation. Twelve of 329 patients (3.6%) underwent open reduction, because closed reduction failed. Overall, 12 of 329 patients (3.6%) required subsequent surgery, i.e., 10 patients (3.0%) because of unsatisfactory repositioning and 2 patients (0.6%) due to neurovascular complications. The mean time from hospital admission to surgery was 6.5 h (range: 0.5 h to 27.0 h). 

Closed or open reduction of ScHF was performed under general anesthesia with the patients placed in the supine position. This was followed by fracture fixation by diverging K-wires inserted from the lateral aspect of the distal humerus. Subsequently, each patient received a long-arm plaster cast.

### 2.4. Ethical Approval

Ethical approval (PCN/022/KB/11/21) was waived by the local Ethics Committee of the Silesian Medical University of Katowice, Poland, because of the retrospective nature of the study and the fact that the procedures were part of routine care. 

### 2.5. Statistical Analysis

We analyzed the data using R (version 3.6.1 operating on x86_64, linux-gnu platform). Differences between groups were assessed using multi-way ANOVA verified by non-parametric Kruskal–Wallis and adonis tests form the vegan R-package. All predictions of ANOVA were in agreement with non-parametric results, thus confirming the robustness of parametric methods. A *p*-value of <0.05 was considered statistically significant. 

## 3. Results

### 3.1. Patient Demographics and Number of Patients per Gartland Type at Baseline

We included 329 patients (140 girls and 189 boys) in this analysis. The mean age of the children was 7.2 years (CI: 6.89, 7.45), and the mean Gartland score amounted to 2.86 (CI: 2.75, 2.97).

The catchment area of the Department of Pediatric Traumatology and Orthopedics, Upper Silesian Child Centre, Katowice, comprised the population of the Silesian county (4.5 mill. inhabitants) with 16.8% of the population younger than 18 years of age. Table 1 shows the patients’ demographics. 

Table 2 demonstrates the mean and variation of Gartland scores and patients’ ages for the total study population.

The mean age of girls (*n* = 140) amounted to 6.9 years (range: 1.5 years—14 years). Boys (*n* = 189) were 7.4 years old on average (range: 6 months—16 years). The mean ages of girls and boys did not differ significantly. 

Figure 1 shows the total number of male and female patients per Gartland type (I to IV) at baseline. The mean Gartland score was 2.9 for both girls and boys. Thus, the mean Gartland score was not influenced by gender.

### 3.2. Relationship between the Incidence of Neurovascular Lesions and the Gartland Score

Overall, 44 of the 329 patients (13.4%) exhibited neurovascular lesions (group 1) at baseline. Group 2 consisted of 285 patients (86.6% of the 329 patients) without neurovascular lesions (Table 3). Table 3 shows the numbers of patients with and without neurovascular lesions by Gartland type. 

Notably, no patient developed late-presenting neurovascular complications or compartment syndrome during the 24 h in-patient observation period. 

The mean Gartland scores were higher in the group of patients with neurovascular lesions (group 1) than in those without neurovascular lesions (group 2). Thus, there was a positive association between the incidence of neurovascular lesions and the Gartland score at baseline (*p* = 0.045; Table 4).

Fractures of low Gartland types were associated with fewer neurovascular lesions. In the patient group with neurovascular lesions (group 1), only one patient with Gartland type I fracture and two patients with Gartland type II fractures suffered accompanying nerve injuries (Table 5). The incidence of vascular injuries peaked in patients with Gartland type IV fractures, while nerve lesions occurred most frequently in patients with Gartland type III and IV fractures (Table 5).

#### Types of Nerve Injuries Associated with ScHF

Nerve lesions without accompanying vascular injuries occurred in 32 of 329 patients (9.7%) (Table 6). 

Eight children (2.4%) exhibited combined neurovascular injuries (Table 5). Among these, one child suffered anterior interosseous nerve injury and brachial artery thrombosis, which required surgical treatment. Another two children presented with anterior interosseous nerve injury combined with brachial artery injury, and five patients suffered median nerve injury accompanied by brachial artery injury. 

We did not observe any posttraumatic compartment syndrome or posttraumatic osteomyelitis.

### 3.3. Relationship of the Patient Age and the Presence of Neurovascular Lesions

The mean age of ScHF patients with neurovascular lesions (group 1; *n* = 44) was 8.2 years (CI: 7.32, 9.02). In patients who did not suffer any neurovascular complications (group 2; *n* = 285), the mean age was 7.0 years (CI: 6.72, 7.31). Thus, patients without neurovascular lesions were significantly younger than those who suffered neurovascular lesions (*p* = 0.04; Table 4, Figure 2). 

The mean age of patients increased with the increasing Gartland type (Table 7).

### 3.4. Relationship of Neurovascular Lesions, Gartland Type, and Patient Gender

The mean age of the girls with neurovascular complications amounted to 7.6 years (range: 2–14 years), and their mean Gartland score was 3.6. The mean age of the girls without neurovascular complications was 6.8 years (range: 1.5–14 years), with a mean Gartland score of 2.8. In the group of boys with neurovascular complications, the mean age was 8.5 years (range: 3–15 years), and the mean Gartland score reached 3.4. The mean age of the boys without neurovascular complications was 7.2 years (range: 6 months–16 years), and their mean Gartland score reached 2.8. There was no significant influence of gender on the incidence of neurovascular lesions in children with ScHF.

### 3.5. Relationship of Fracture Laterality (Right or Left) on the Gartland Score

Extension-type fractures constituted 99% of all fractures. The left upper extremity was affected in 218 of 329 patients (66.3%; Figure 3). The mean Gartland score of fractures of the left humerus was 2.9, and the mean age of the affected children was 7.3 years (range: 1.5–15 years). The mean Gartland score of fractures of the right humerus was 2.8, and the patients’ mean age was 7.0 years (range: 6 months–16 years). Thus, the laterality of the fracture did not influence the Gartland score, nor was there any significant age difference between patients with left-sided or right-sided humerus fractures (Figure 3).

### 3.6. Outcome of Nerve Lesions and Vascular Injuries

In 30 of the 32 children (93.8%) with nerve injuries at baseline, nerve lesions resolved spontaneously within four weeks after the reduction and stable fixation of ScHF. In one patient suffering from Gartland type III ScHF, open reduction was conducted initially. The surgical release of the ulnar nerve was performed two months after the injury, because ulnar nerve function did not improve. In an additional patient with Gartland type IV ScHF, we undertook the surgical release of the median nerve after four weeks due to missing signs of nerve recovery. In these two patients, nerve function recovered after the averages of 100 and 146 days, respectively. 

The 20 children who presented with absent forearm pulses and the diminished oxygen saturation of the ipsilateral hand (Table 5) underwent closed fracture reduction. After the reduction and stabilization of the fracture, we re-evaluated forearm pulses and the oxygen saturation of the hand and conducted the Doppler sonography of the ipsilateral hand and forearm arteries. In 17 of these patients, perfusion improved after closed reduction, while the surgical revision of the brachial artery to treat thrombosis was necessary in three patients. One of these children sustained an open supracondylar fracture (Gustilo-Anderson grade IIIc).

At follow-up at least one year after the injury, we noted no differences in limb growth. One child complained of intolerance to cold temperatures. The duplex sonography revealed a reduced flow (<50%) of the ipsilateral forearm arteries in three of the 20 children who initially presented with absent forearm pulses.

## 4. Discussion

In our study, in pediatric patients, the incidence of neurovascular complications was related to the severity of ScHF displacement as rated by the modified Gartland classification [2]. However, nerve injuries occurred predominantly in Gartland types III and IV fractures, while acute vascular injuries were seen mainly in Gartland type IV fractures. We found a higher prevalence of Gartland type I fractures in younger children and a higher incidence of neurovascular complications in older children. The gender and laterality of fractures did not affect the incidence of neurovascular complications.

Vallila et al. reported that 63% of compensation claims for complications in pediatric fractures in Finland between 1990 and 2010 were related to the treatment of ScHF [23].

Early complications including vessel and nerve damage, muscle destruction, and acute compartment syndrome usually appear immediately after suffering ScHF. Varus and, less frequently, valgus deformity, hyperextension malalignment, and restriction of elbow range of motion represent the most common late complications of displaced ScHF in children [2,24].

We observed a mean time from admission to surgery of 6.5 h which is similar to the time interval proposed by Pavone et al. (<8 h) [25]. However, Schmid et al. showed that postponing surgery does not influence the rate of open reductions, incidence of postoperative complications, and overall outcome [26]. 

We performed closed reduction and percutaneous lateral pin fixation in the supine position of the patient. Pavone et al. demonstrated good radiographic outcomes and similar complication rates after closed reduction and percutaneous pinning in Gartland type III fractures treated in either the supine or the prone position of the patient [25].

### 4.1. Incidence of Nerve Lesions

According to the literature, ScHF is complicated by nerve injuries in 5.8% to 14% of pediatric patients [5,6,12,13]. Most nerve lesions associated with ScHF are caused by nerves entrapped within the fracture rather than by sharp bone fragments or K-wires [27]. In our study, the incidence of nerve lesions was related to the degree of fracture displacement and amounted to 9.7% in total. This figure is in line with previously published incidences of nerve injuries in ScHF [5,6,12,13].

In the study reported by Valencia et al. [12], neurovascular complications occurred exclusively in Gartland type III injuries. In our study population, we did, however, observe a small number of neurologic lesions even in Gartland type I (*n* = 1) and type II (*n* = 2) fractures, although the incidence peaked in Gartland type III and type IV fractures.

#### 4.1.1. Types of Nerve Injuries Associated with ScHF

In our study, most neurologic injuries involved the median nerve and anterior interosseous nerve (Table 6). The type of nerve injury is influenced by the fracture type and direction of fracture displacement [28]. 

The median nerve tends to be affected in case of posterolateral displacement [29]. In the literature, the most common neurologic deficit associated with ScHF in children is reported to be neurapraxia which usually resolves spontaneously within two to three months but can take up to six months [12,30]. In accordance with Omid et al., we noted that the most frequent nerve complications related to ScHF in children were median and anterior interosseous nerve injuries [2]. 

#### 4.1.2. Assessment of Neurologic Function

Technically, it is possible to differentiate between anterior interosseous nerve and median nerve palsies by assessing motor function, based on the flexion of the middle-finger proximal interphalangeal joint [30]. However, the assessment of neurologic function before and after surgical treatment in young children may be challenging or even impossible due to the children’s cooperation [6,13,31]. 

There is no significant association between the timing of the surgical procedure and partial or complete nerve recovery [32]. In our study, the recovery of impaired nerve function took an average of four weeks in 93.8% of patients who initially presented nerve injuries. 

### 4.2. Iatrogenic Nerve Injuries

Current studies report an incidence of iatrogenic nerve injury between 3% and 4% [12,13]. It is noteworthy that we did not observe any iatrogenic nerve injuries in our study population. We used the lateral pinning technique that permits the safe and reliable stabilization of Gartland type II, III, and IV ScHF [33]. Pesenti et al. demonstrated that crossed pin constructs are more prone to ulnar nerve injury than lateral pin constructs [34]. Minimal incision over the medial epicondyle in order to identify the bony prominence of the medial epicondyle in case of massive soft tissue swelling can help to reduce the rate of ulnar nerve injuries in crossed pin constructs [34]. Larson et al. reported that crossed pinning is biomechanically more stable than lateral pin constructs but carries a higher risk of ulnar nerve injury during the insertion of the medial pin [35]. Afaque et al. and Pavone et al. demonstrated that there is no significant difference between lateral pinning and crossed pinning in terms of functional outcome, biomechanical stability, and incidence of complications [10,36].

### 4.3. Incidence of Non-Iatrogenic Vascular Injuries

Traumatic, non-iatrogenic vascular injuries in children are rare and mainly involve the brachial artery. 

Surprisingly, a single patient in our subgroup of patients with Gartland type III fractures experienced isolated arterial injury, while two patients in this subgroup presented with combined nerve and arterial lesions. In our study, most isolated arterial lesions occurred in patients with Gartland type IV fractures (*n* = 11), while six patients in this subgroup exhibited combined neural and arterial injuries. Our findings are in accordance with the observation of Garbuz et al. who described that the incidence of concomitant nerve lesions in patients suffering from ScHF complicated by absent radial pulse is 60% [37].

### 4.4. Combined Vascular and Neurologic Lesions

Louahem et al. [38] noted the possibility of combined neurovascular complications in patients with ScHF. The therapeutic approaches in these patients were the same as in isolated nerve or vessel injuries. In our patients with ScHF, we observed neurovascular complications mainly in Gartland type IV (six patients) and type III (two patients) ScHF, confirming our previous observation of a higher incidence of neurovascular complications in ScHF with significant displacement and instability.

### 4.5. Study Limitations and Strengths

#### 4.5.1. Study Limitations

The main study limitation was the retrospective study design and lack of standardization of evaluations. Furthermore, various medical teams attended to our patients because of the prolonged study period (2004–2019), and diagnostic procedures may have changed in the course of the 15-year period. 

#### 4.5.2. Study Strengths

Our study describes the functional outcomes of a large number of pediatric ScHF patients who were treated by surgical stabilization. Operations were performed by different surgeons, and therefore, our results may be generalized. We were able to confirm the association between the more marked fracture displacement (i.e., higher Gartland type) and the incidence of neurovascular complications.

We demonstrated a good recovery of impaired neurovascular function after closed reduction and stable retention by K-wires inserted from the lateral aspect of the upper arm.

## 5. Conclusions

The incidence of neurovascular complications was related to the degree of ScHF displacement as classified according to Gartland. Vascular complications mainly accompanied Gartland type IV ScHF, whereas nerve lesions occurred in Gartland type III and IV ScHF.For the treatment of displaced ScHF, we recommend closed reduction and stabilization by K-wires inserted percutaneously from the lateral aspect of the distal humerus. If the impaired perfusion of the forearm persists after fracture reduction and stabilization or if complete nerve paralysis or iatrogenic nerve lesion develops, surgical treatments of these neurovascular complications should be considered.

## Figures and Tables

**Figure 1 children-09-00308-f001:**
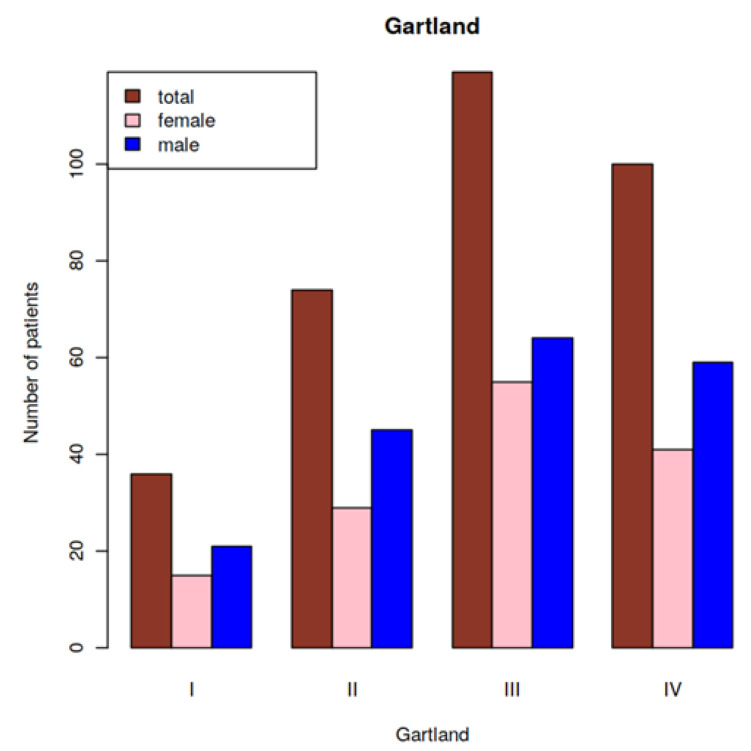
Number of patients (total, male, and female) by Gartland type.

**Figure 2 children-09-00308-f002:**
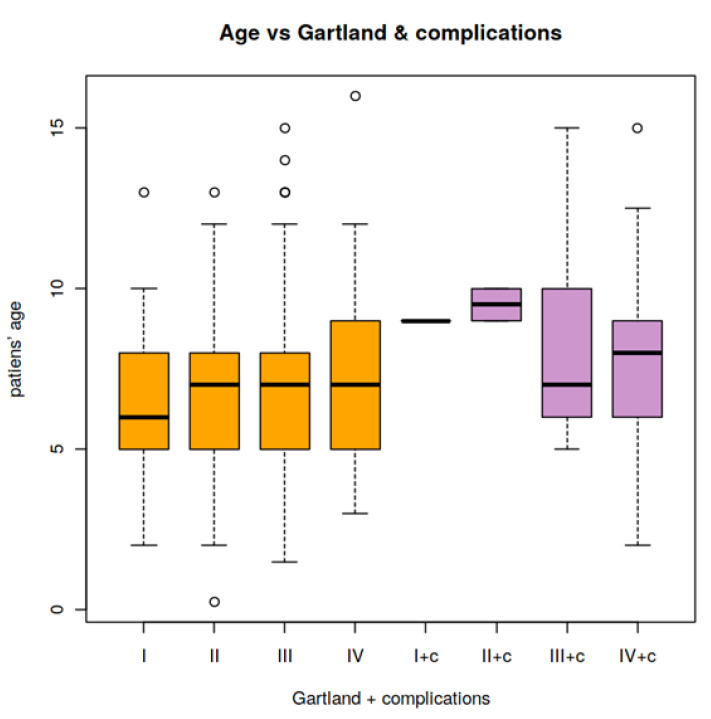
Ages of patients without (orange) and with (purple) neurovascular lesions by Gartland type. c, complications.

**Figure 3 children-09-00308-f003:**
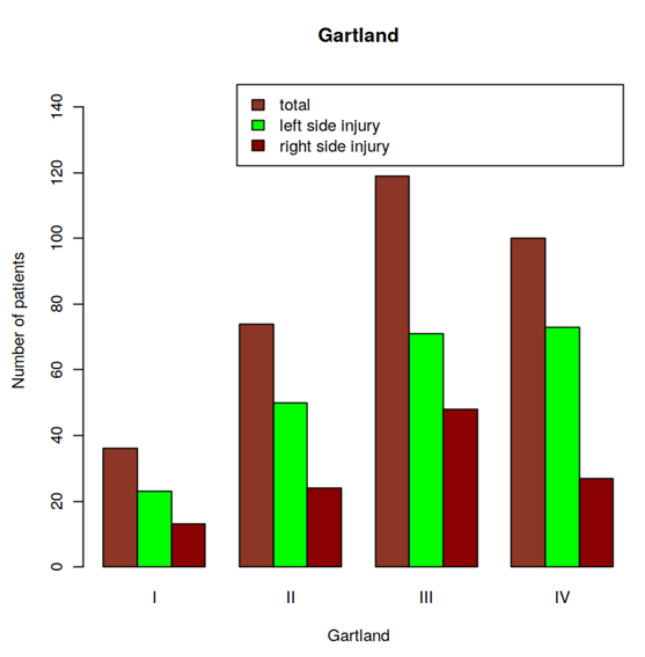
Number of patients per fracture laterality and Gartland type.

**Table 1 children-09-00308-t001:** Patients’ demographics (*n* = 329).

Gender (female/male; %)	140 (42.6%)/189 (57.4%)
Age (years; CI)	7.2 (CI: 6.89, 7.45)
Ethnicity (*n*; %)	Caucasian (326; 99.1%); Asian (2; 0.61%); African (1; 0.3%)

**Table 2 children-09-00308-t002:** Mean values and variation of Gartland scores and patients’ ages at baseline.

All Patients (*n* = 329)	Mean and CI	Variation and CI
Gartland score	2.86 CI = 2.75–2.97	0.95 CI = 0.82–1.11
Age (years)	7.2 CI = 6.89–7.45	6.76 CI = 5.84–7.93

**Table 3 children-09-00308-t003:** Numbers of patients with and without neurovascular lesions per Gartland type.

Number of Patients	Gartland Type			
	I	II	III	IV
Total (*n* = 329)	36	74	119	100
With neurovascular lesions (group 1; *n* = 44)	1	2	14	27
Without neurovascular lesions (group 2; *n* = 285)	35	72	105	73

**Table 4 children-09-00308-t004:** Mean Gartland scores and variations and mean patients’ ages for the patients with and without neurovascular lesions.

	Patients with Neurovascular Lesions (Group 1, *n* = 44)		Patients without Neurovascular Lesions (Group 2, *n* = 285)		*p*-Value
	Mean and CI	Variation and CI	Mean and CI	Variation and CI	
Gartland score	3.52 CI = 3.31–3.73	0.48 CI = 0.33–0.78	2.75 CI = 2.64–2.87	0.94 CI = 0.81–1.12	0.045
Age (years)	8.2 CI = 7.32–9.02	7.81 CI = 5.33–12.54	7.0 CI = 6.72–7.31	6.45 CI = 5.51–7.66	0.04

**Table 5 children-09-00308-t005:** Number of patients with neurovascular lesions per Gartland type.

Number of Patients with Neurovascular Lesions	Gartland Type			
	I	II	III	IV
Total (*n* = 44)	1	2	14	27
Nerve lesions (*n* = 24)	1	2	11	10
Vascular lesions (*n* = 12)	0	0	1	11
Combined neurovascular injuries (*n* = 8)	0	0	2	6

**Table 6 children-09-00308-t006:** Types of nerve lesion and Gartland types of nerve injuries not accompanied by vascular lesions.

Type of Predominant Nerve Lesion	Gartland Type			
	I	II	III	IV
Median nerve lesion	1	2	6	9
Anterior interosseous nerve lesion	0	0	3	4
Ulnar nerve lesion	0	0	3	3
Radial nerve lesion	0	0	1	0

**Table 7 children-09-00308-t007:** Mean value and variation of patients’ ages per Gartland type.

All Patients (*n* = 329)	Gartland Type			
	I	II	III	IV
Mean age and CI (years)	6.4 CI = 5.63–7.26	6.9 CI = 6.39–7.5	7.4 CI = 6.87–7.85	7.4 CI = 6.86–7.91
Variation	5.74 CI = 3.78–9.77	5.73 CI = 4.25–8.16	7.31 CI = 5.76–9.61	7.12 CI = 5.49–9.6

## Data Availability

The data presented in this study are available on request from the first author. The data are not publicly available due to hospital guidelines.

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
