# Peer review of "Supracondylar Fractures of the Humerus: Association of Neurovascular Lesions with Degree of Fracture Displacement in Children—A Retrospective Study"

_children, 2022, doi:10.3390/children9030308_

Round 1
Reviewer 1 Report
The subject of the article is well written and relevant, and the discussion presents a comprehensive review of the literature to support your 15+ year chart analysis. The retrospective chart analysis adds clinical significance to the review of current treatments.
- Adding a chart of patient demographics would be helpful to better define your population.
- Please check line page 4 line 147, do you mean to reference Table 3?
- Page 5, section 3.2.1 presents some important findings and speaks to your aim/contributes to disproving your null hypothesis, should merit a table for nerve injuries by Gartland type, especially as you include the unstable Gartland IV, which few articles do.
- Kudos on your track record for lateral pinning.
Author Response
Dear reviewer,
Please see the attached response to your review report which helped us to improve our manuscript.
Kind regards,
Johannes Mayr

Reviewer 2 Report
The authors of this manuscript intend to evaluate the possible relationship between fracture displacement in supracondylar humerus fractures and neurovascular injury.
Overall, the topic is very topical and quite interesting. The authors have put a lot of effort into the above-mentioned correlation. However, I have the impression that the text as a whole is far too detailed. The reader has the impression that he is reading a review article on the general treatment of supracondylar humerus fractures. However, according to the hypothesis, this was obviously not the intention of the authors. Thus, a comprehensive review of the article must be carried out. On the one hand, the text must be drastically shortened. Secondly, the focus should be solely on evaluating the relationship between fracture displacement in supracondylar humerus fractures and neurovascular injuries. The reader does not expect general therapy concepts and suggestions for the treatment of these fractures. Surgical techniques should not be the subject of this lecture, nor is the adulation of one's own surgical method without describing it in detail clearly out of place here.
In particular, the entire discussion must be refocused in this context. Furthermore, 80 citations are clearly too many, as this is not intended to be an overview article.
Author Response
Dear reviewer,
Many thanks for your review report.
Please see the attached response to your review report.
Kind regards,
Johannes Mayr

Reviewer 3 Report
Many thanks to the authors for their efforts in this study. Unfortunately, there is no innovation in the research content of the article, and the research conclusions have been reported in a number of similar studies earlier.
Author Response
Dear Reviewer 3,
We revised the English of our manuscript with the support of Silvia Rogers, PhD, director of MediWrite Comp. Basel, Switzerland. We added further information (Table 1 and Table 6) and shortened our article and refocused the discussion section and deleted many references.
Please see the attached response to reviewers file.
Kind regards,
Johannes Mayr

Round 2
Reviewer 2 Report
The text was edited sufficiently according to my suggestions.
Reviewer 3 Report
Many thanks to the authors for their efforts in this study. Unfortunately, there is no innovation in the research content of the article, and the research conclusions have been reported in a number of similar studies earlier.
